# A Narrative Review about Autism Spectrum Disorders and Exclusion of Gluten and Casein from the Diet

**DOI:** 10.3390/nu14091797

**Published:** 2022-04-25

**Authors:** Pablo José González-Domenech, Francisco Diaz-Atienza, Luis Gutiérrez-Rojas, María Luisa Fernández-Soto, Carmen María González-Domenech

**Affiliations:** 1Department of Psychiatry, University of Granada, Health Technology Park, Av. de la Investigación, 11, 18071 Granada, Spain; pgdomenech@ugr.es (P.J.G.-D.); fdatienza@ugr.es (F.D.-A.); gutierrezrojas@ugr.es (L.G.-R.); 2Psychiatry Service, Hospital San Cecilio, Health Technology Park, Av. de la Investigación, 11, 18016 Granada, Spain; 3Department of Medicine, University of Granada, Health Technology Park, Av. de la Investigación, 11, 18071 Granada, Spain; mlfernan@ugr.es; 4Department of Microbiology, University of Málaga, Louis Pasteur Avenue, 29010 Málaga, Spain

**Keywords:** diet, gluten, casein, autism spectrum disorders

## Abstract

Objective: Autism spectrum disorders (ASDs) appear in the early stages of neurodevelopment, and they remain constant throughout life. Currently, due to limitations in ASDs treatment, alternative approaches, such as nutritional interventions, have frequently been implemented. The aim of this narrative review is to gather the most relevant and updated studies about dietary interventions related to ASDs etiopathogenesis. Results: Our literature search focused on the gluten- and casein-free (GFCF) diet. The literature found shows the inexistence of enough scientific evidence to support a general recommendation of dietary intervention in children with ASD. Protocols and procedures for assessing risk and safety are also needed. Future lines: Prospective and controlled research studies with larger sample sizes and longer follow-up times are scarce and needed. In addition, studies considering an assessment of intestinal permeability, bacterial population, enzymatic, and inflammatory gastrointestinal activity are interesting to identify possible responders. Besides brain imaging techniques, genetic tests can also contribute as markers to evaluate the comorbidity of gastrointestinal symptoms.

## 1. Introduction

Autism belongs to a heterogeneous and complex neurodevelopmental disorders group called autism spectrum disorder (ASD). ASD also comprises several other conditions, previously considered the following independent disorders: Asperger syndrome, childhood disintegrative disorder, and pervasive developmental disorder [1]. All these conditions share the following two main signs: difficulty with social interaction and communication, as well as repetitive and stereotyped patterns of behavior, interests, and actions [1]. In addition, there exist other peripheral symptoms associated with the clinical presentation of ASD, such as sensory and perception abnormalities [2], motor skills problems [3], and digestive disorders [4].

The prevalence of ASD has recently increased, affecting approximately 0.62–0.70% of the European population [5]. Not only an improvement in the diagnosis of ASD explains this increasing prevalence, but also the other following factors: wider nosological criteria, better healthcare, and changes in case definition [6]. Furthermore, recent studies also point to environmental factors as a cause of its current growth in prevalence, and among them, the dietary ones [7]. Therefore, it is well accepted that the etiology of ASD involves a causal interaction between environmental and genetic issues.

The multifactorial origin of ASD and the various influences over the first stages of neurological development [8] address the interventions from several areas with the final goal of reducing disabilities or enhancing giftedness and improving the quality of life for ASD patients. One of these areas concerns dietary approaches because of the suggested relationship between ASD and gastrointestinal symptoms and disorders such as celiac and inflammatory bowel diseases [8,9,10,11].

This review has gathered all the updated studies exploring the association of ASD, gastrointestinal symptoms, and nutrition and exposes our findings as a narrative overview. We pay special attention to dietary approaches and interventions related to gluten- and casein-free diets (GFCF) in autism.

## 2. Nutritional and Dietary Interventions for Autism Spectrum Disorder

### Background

Nutritional and dietary interventions in autism are currently gaining popularity (Figure 1). Notorious improvement in symptoms of ASD has been reported after suspension of certain nutrients [12,13,14,15,16] and/or use of nutritional supplements such as vitamins, minerals, amino acids, fatty acids, prebiotics and probiotics, and a ketogenic diet and GFCF regimes, among others [17,18,19,20,21,22,23].

The gut microbiota plays a key role in affecting eating behavior, and the so-called microbiota-gut-brain axis might explain the diverse metabolic and nutritional profiles found in children with ASD. Dohan was the first author to propose the influence of nutrition on psychiatric disorders. Thus, he observed the improvement of schizophrenia symptoms by eliminating foods containing gluten and casein [24]. Panksepp suggested an association between behavioral disorders in ASD and excess opioid receptor agonists [25]. Both theories represented the starting point to study the opioid effect of peptides from gluten and casein and how dairy products containing them might impact autism spectrum conditions [12,26,27,28]. The underlying mechanism of the above-exposed hypotheses is the insufficient hydrolysis of proteins from the diet, besides the abnormally increased intestinal permeability and absorption of peptides in children with autism [29]. Therefore, gluten and casein may provide ‘opioid-like’ peptides, which would reach blood flow and, by systemic dissemination, cross the blood–brain barrier (Figure 2). These circulating peptides might trigger a systemic inflammatory response [30,31] and, finally, involve the central nervous system. Such a result could be toxic when it appears in the early stages of neurodevelopment [8] and worsens autistic symptoms [32].

Moreover, the limited success in the treatment of ASD negatively impacts parental stress and decision making, pushing families to alternative interventions, and among them, the nutritional ones, often without medical supervision [33,34,35]. Thus, a recent study shows a high percentage of parents (up to 33%) hiding information about the supplemented diet of their children with ASD from clinicians [36].

## 3. Combined Gluten- and Casein-Free Diet (GFCF Diet)

We provide updated coverage of literature gathered regarding the GFCF diet in Table 1. Ten out of 15 selected publications possess a cohort design; 3 of them were crossover studies, and only 1 was a case-control study. All the interventions included were addressed between 1 week and 2 years [16,37]. The mean age of study participants ranged from 2 to 18 years [38,39]. In addition, the comparison groups, when considered, received a normal diet in 10 studies [12,13,14,16,38,39,40,41,42,43], a GFCF diet with a dietary supplement containing gluten and/or casein in four of them [37,44,45,46], and a diet with low sugar content in one study [47].

### 3.1. Positive Findings for a GFCF Diet

The first reports about the outcomes of GFCF in ASD arose during the early 90s, with Knivsberg et al. studies [41,42]. These Nordic researchers followed up on the behavior of a group of 15 ASD patients on a GFCF diet for up to 4 years. They found an improvement in autistic actions, especially after the first year of intervention, and decreased epileptic seizures and urinary peptide levels from gluten and casein metabolism. A few years later, the team headed by Whiteley and Shattock in the United Kingdom published similar positive results (except for peptides in urine) with a GFCF diet and for a group of 22 children with ASD (*n* = 18), dyspraxia (*n* = 2), and semantic-pragmatic disorder (*n* = 2) [43]. However, both studies showed important methodological limitations (non-randomized and open design). Thus, Knivsberg and Reichelt enhanced the previous studies in 2002 [12] with a single-blind, randomized, and controlled clinical trial; 20 children diagnosed with ASD were followed up for 1 year, half of them with GFCF and the rest with a normal diet with foods containing gluten and casein. The GFCF group improved their communication, behavior, and social interaction skills.

Finally, all the authors mentioned above joined together in a common study called ScanBrit because of their both origins (Scandinavian and British). It consisted of a single-blind, randomized clinical trial with 2 years of follow-up and a wider sample population (*n* = 72). ScanBrit study showed a significant benefit of GFCF diet in the neurological development and behavioral disorders within the 12 months of follow-up but remained a plateau effect after 1 year [16]. In addition, the ScanBrit study presented individual variability, so patients were subsequently divided into best- and non-responders to such dietary intervention. Responders were afterward defined as those with clinically significant positive change scores for hyperactivity and inattention behavior [15]. Furthermore, the median age was also considered a predictor factor of success in responders, so children between 7 and 9 years old were in the best age range to succeed with a GFCF diet. The ScanBrit study is a world reference in dietary interventions based on the GFCF diet because of its wider sample size and longer follow-up period. Nevertheless, a placebo design and monitoring concomitant treatments are absent, determining a lack of scientific robustness and the chance of being a recommendation in the field [48].

Besides the ScanBrit study and its remarkable findings, plenty of other researchers have found positive results with a GFCF diet for ASD patients, especially in the improvement of language communication skills, social interaction, stereotyped behavior, and motor coordination, as well as a decrease in hyperactivity, self-destructive behavior, seizure activity, and gastrointestinal disorders [13,14,15,16,17,41,42,47]. Other studies also show positive results, but in smaller cohorts or even individual cases [10,49,50]. In addition, other assessments of the GFCF diet in ASD were only based on the opinions of close relatives. Thus, Pennesi and Klein, after following up with 293 children with ASD following a diet without gluten and casein proteins, asked their parents; the best results were obtained for patients following this diet for more than 6 months and showing previously any gastrointestinal and allergic symptoms [51].

### 3.2. Non-Significant Improvement with GFCF Diet

Although the results about a GFCF diet shown in the previous section appear consistent and helpful for ASD patients, we also found recent literature (mainly from the USA) regarding negative outcomes and non-significant improvement results [37,39,44,46,47,48,52,53].

The first study providing lack of any significant effect of gluten and casein removal from the diet in children with autism was from North America in 2006 [39]. This research possessed several methodological strengths compared to the ScanBrit study, as follows: a double-blind crossover design and a retrospective observation by videotaping unscheduled sessions (15 min each) recording ASD children’s interactions with their primary caretaking parent. So, they found no statistically significant improvement with a GFCF diet, even though some parents hold the contrary view. However, the Elder et al. study showed the following several limitations: sample size (*n* = 15) and dietary intervention period (6 weeks with a GFCF diet, followed by 6 weeks with a normal diet). A few years later, Johnson et al. performed a single-blind clinical trial, finding no statistically significant difference between GFCF and the control arm [47]. Nevertheless, their sample size (*n* = 22) and follow-up period (12 weeks) are similar to the Elder et al. study and far lower than the ScanBrit’s figures.

In 2015, two studies describing a lack of significant results on this topic were published. The first one was a double-blind controlled placebo design, comprising 74 ASD patients supplemented with gluten and casein for a week; after that, the authors found no significant improvement neither in behavioral and gastrointestinal symptoms nor urinary peptide levels [37]. The second clinical trial published that year was led by Navarro’s team, with a similar design as well as gluten and casein supplementations, but during the 4 weeks and with a smaller sample size (*n* = 12), they found no significant improvement, neither in behavior nor in intestinal permeability levels [46].

Doctor Susan Hyman et al. published their results in 2016 [44]. She performed a double-blind, randomized, and controlled placebo clinical trial to assess the efficacy and safety of a GFCF diet for 14 children diagnosed with ASD. Hyman et al. chose a narrow range of age (3–5 years old) for a larger parenteral dietary control at such an age. Moreover, another study also supported a greater potential for responding to any dietary intervention and treatment for this age range [54]. Hyman’s study comprised the following three stages: implementation, intervention, and maintenance, with different follow-up periods (6 weeks for the former and 12 weeks for the last two ones). During the implementation phase, a nutritionist established and monitored the GFCF diet and attended each participant’s home at least twice during the period. Additionally, dietary advice for parents was weekly provided via telephone. Then, an intervention phase consisted of randomized supplying snacks containing only gluten or casein, both of them or none of them (placebo), once a week. During the implementation and intervention phases, the researchers weekly measured the nutritional and behavioral status of participants. The weekly basis of those measures was well-founded in previous survey data confirming secondary effects of gluten and casein appearing after one week and in clinical studies of adverse reactions to foods and food additives [51,55]. Finally, the participants’ parents made a completely free decision about remaining, discontinuing, or modifying the GFCF diet during the maintenance stage. At the end of this phase, the nutritional and behavioral status of participants were also assessed. Although Hyman’s study showed safety and tolerance for a GFCF diet when combined with nutritional expert advice, the results were not statistically significant for autistic symptoms and related behavioral disorders [44]. Moreover, despite its small sample size (only 14 participants), this research possesses several strengths, such as the study design, and mainly the strict inclusion criterion requiring a steady status for the previous 4 months and the absence of any pharmacological treatment during the study.

Regarding non-significant results in ASD children with a GFCF diet, our team recently published the outcomes from two single-blind, crossover, and randomized clinical trials [38,40]. Our research methodology was the same as used in other studies about the GFCF diet in ASD children and young adults [39,53], but the strong point was that each participant behaved as a case and as a control at the same time. Thus, this design reduced inter-subject variability and allowed researchers to work with smaller sample sizes, which is advantageous because of the difficulties in recruiting patients for this kind of study [48]. The first study assessed 28 children and youths for 6 months, half of the time following a GFCF diet and the other half with a normal diet [38]. The second study enrolled a larger sample size (*n* = 37) and a longer period of follow-up (12 months) [40]. In addition, this last clinical trial considered gastrointestinal comorbidities and potential secondary effects from gluten- and casein-free diets. Neither of them found statistically significant differences among patients following a GFCF or a normal diet, nor even in gastrointestinal symptoms or nutritional status, and anthropometric variables.

Summarizing, all the studies with no significant results are double-blind designed [37,39,44,46,47,53]. The main disadvantage of this design for dietary intervention considering the GFCF diet and ASD together is the small sample size and the short follow-up period, all of them because of the expensive costs and the methodological difficulty [56]. In contrast, the opposite features mostly characterize the following positive results: larger sample size and longer follow-up period, but they are neither double-blinded nor standardized to assess adherence. This kind of research is mainly from the Scandinavian and British mentioned teams [16,41,42,43].

Regarding sample size, a minimum value of 30 participants exists to assess the influence of a GFCF diet on autism. This figure was obtained by Knivsberg et al. [12], assuming α and β errors of 0.05 and 0.90, respectively, with an estimated probability of improvement of 60% for the GFCF group compared to the 10% expected in the control group (normal diet), for a 1:1 ratio. Thus, when group reassignments are allowed, as in the ScanBrit study, there is more flexibility and closeness to real life, so dropout rates decrease. Consequently, this scenario is the best for researching disorders with low enrolment options, such as ASD [57].

Concerning the intervention period, 3 months is the minimum duration to observe a positive result with a GFCF diet. This follow-up time is established based on the remaining activity of gluten and its bioproducts in celiac disease patients after eliminating this protein from the diet [58]. Nevertheless, ScanBrit and other sources lying on parents’ point of view suggest an even longer period of intervention, up to 6 months [16,51].

In addition, age range also seems to be an important response factor to dietary intervention, with the best results for younger ASD patients [16,59]. However, the basis for such an observation remains unclear, arguing, on the one hand, brain plasticity and neuronal maturation, and on the other hand, a simple coincidence because of diagnostic instability at that age range.

Dietary interventions in ASD patients are implemented in almost half of children with ASD [49,60,61]. However, considering all these factors and the above-exposed results, several authors suggest thoughtfulness in the general adoption of these interventions [39,48,52,62,63,64,65,66]. Moreover, some researchers only recommend a GFCF diet for cases of previous gluten and/or casein allergy or intolerance [48,52,63].

## 4. Conclusions

This review has assessed the different dietary and nutritional interventions implemented in ASD patients. Moreover, the physiopathological basis of such therapies, besides clinical, genetic, and inflammatory biomarkers suggestive of answers, has been thoroughly reviewed.

Currently, there is not enough scientific and clinical knowledge to recommend such interventions to all ASD patients. Hence, further randomized clinical trials are needed, comprising a longer follow-up period and a double-blind design, including a placebo. Other assessments should be performed to identify the following potential candidates as successful responders: enzymatic and inflammatory intestinal activity, intestinal permeability measures, microbiome studies, evaluation of other comorbidities with gastrointestinal symptoms, genetic and neuroimaging tests, etc.

## Figures and Tables

**Figure 1 nutrients-14-01797-f001:**
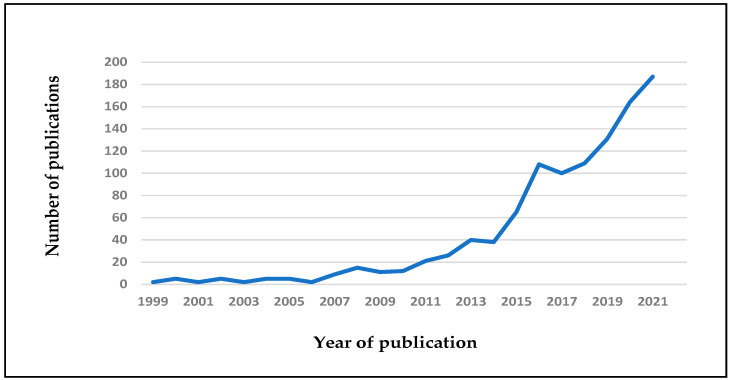
Trend of publications in English related to nutrition and autism spectrum disorders (ASD), from 1999 to 2021. Data from Pubmed, with the following keywords: nutrition, autism spectrum disorder; date of search 5 April 2022.

**Figure 2 nutrients-14-01797-f002:**
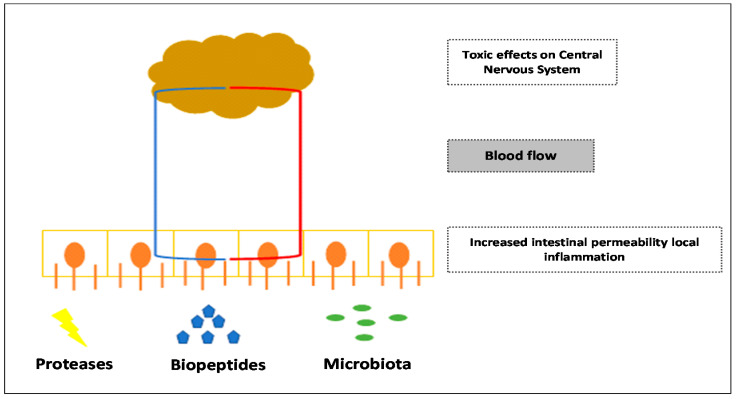
Microbiota-gut-brain axis in autism spectrum disorders.

**Table 1 nutrients-14-01797-t001:** Main studies (in chronological order of publication) concerning the efficacy of gluten-free, casein-free diets in autism spectrum disorders.

Author	Country	Sample Characteristics	Study Design	Follow-up Period	Assessment Methods	Outcomes
Knivsberg et al., 1990 [41]	Norway	15 ASD Non-controlled 6–14 years old	Cohorts GFCF	12 months	DIPAB, C-RAVEN Parent/caregiver surveys Levels: Urinary peptides	Behavioral improvement Seizure activity decrease Urinary peptides decrease
Knivsberg et al., 1995 [42]	Norway	(idem: Knivsberg et al., 1990) [41]	(Cont. Knivsberg et al., 1990) [41]	4 years	(Cont. Knivsberg et al., 1990) [41]	Behavioral improvement, less after one year
Lucarelli et al., 1995 [45]	Italy	36 ASD 20 non-ASD controls 8–13 years old	Phase I (cohorts): GFCF Phase II (double-blind): Suppl. casein	Phase I: 2 months Phase II: 2 weeks	ERC Levels: antibodies in serum	Behavioral improvement and antibodies decrease after Phase I Inconclusive results after Phase II
Whiteley et al., 1999 [43]	UK	22 ASD 11 ASD controls 3–8 years old	Cohorts Experimental arm: GFCF Controls: Normal diet/reintroduction gluten	5 months	ERC, K-ABC Parent/caregiver surveys Levels: Urinary peptides	Behavioral improvement Non-significant urinary peptides decrease
Knivsberg et al., 2002 [12]	Norway	10 ASD 10 ASD controls 5–10 years old	Single-blind, randomized Experimental arm: GFCF Control arm: Normal diet	12 months	DIPAB, ITPA, LIPS, Reynell–Spark test, TOMI Parent/caregiver surveys	Behavioral, communication and social interaction improvement
Elder et al., 2006 [39]	USA	15 ASD 2–16 years old	Crossover, double-blind, randomized 6 weeks GFCF + 6 weeks normal diet (without washout period)	3 months	CARS, ECOS Parent/caregiver surveys Levels: Urinary peptides	Non-significant improvement in neither behavioral nor urinary peptides
Whiteley et al., 2010 [16]	UK Norway Denmark	38 ASD 34 ASD controls 4–10 years old	Single-blind, randomized Experimental arm: GFCF Control arm: Normal diet	24 months	ADHD-IV, ADOS, GARS, VABS	Improvement after 12 months; plateau effect after 24 months; hyperactivity enhancement in a range of ages between 7–9 years
Johnson et al., 2011 [47]	USA	8 ASD 14 ASD controls 3–5 years old	Single-blind, randomized Experimental arm: GFCF Controls: Low sugar diet	3 months	ADOS, CBCL, MSEL Secondary effects questionnaire 24 h reminder	Non-significant improvement in behavior nor secondary effects
Pusponegoro et al., 2015 [37]	Indonesia	38 ASD 36 ASD controls 4–6 years old	Double-blind, randomized, placebo: Suppl. gluten + casein vs. rice	1 week	PDDBI, GSSI Levels: Urinary peptides	Non-significant improvement in neither behavioral, gastrointestinal symptoms, nor urinary peptides
Navarro et al., 2015 [46]	USA	6 ASD 6 ASD controls 4–7 years old	Double-blind, randomized, placebo: Suppl. gluten + milk powder vs. whole rice flour	4 weeks	ABC, CBCL, CPRS-R, SCQ Levels: intestinal permeability (lactulose and mannitol)	Non-significant improvement in neither behavioral nor intestinal permeability levels
Hyman et al., 2016 [44]	USA	14 ASD 3–5 years old	Phase I (cohorts): GFCF Phase II (double-blind, randomized, placebo): suppl. gluten vs. suppl. casein vs. suppl. gluten + casein vs. Placebo Phase III (maintenance): GFCF	Phase I: 6 weeks Phase II: 3 months Phase III: 3 months	CPRS; RFRLRS Sleep diaries Secondary effects	Non-significant improvement in neither behavioral, autistic nor physiological symptoms Non-existing secondary effects
Ghalichi et al., 2016 [14]	Iran	40 ASD 40 ASD controls 4–16 years old	Single-blind, randomized Experimental arm: GFCFControl arm: Normal diet	6 weeks	GARS: ROME III	Behavioral and intestinal permeability levels improvement
El-Rashidy et al., 2017 [13]	Egypt	15 ASD 30 ASD controls 3–8 years old	Case-controls, three similar size arms (*n* = 15): GFCF vs. ketogenic diet vs. normal diet	6 months	ATEC, CARS 24 h reminder	Behavioral improvement in GFCF and ketogenic arms, regarding control arm
González-Domenech et al., 2019 [38]	Spain	37 ASD 2–18 years old	Crossover, single-blind, randomized: 6 months GFCF + 6 months normal diet No washout period	12 months	ATEC, ABC, ERC 24 h reminder	Non-significant differences in neither behavior nor urinary peptides
González-Domenech et al., 2020 [40]	Spain	28 ASD 3–16 years old	Crossover, single-blind, randomized 3 months GFCF + 3 months normal diet No washout period	6 months	ATEC, ABC, ERC Food-frequency questionnaire	Non-significant differences in neither behavior nor urinary peptides

Abbreviations: ABC: Aberrant Behavior Checklist; ADHD-IV: Attention-Deficit Hyperactivity Disorder-IV rating scale; ADOS: Autism Diagnostic Observation Schedule; ASD: Autism Spectrum Disorders; ATEC: Autism Treatment Evaluation Test Questionnaire; CARS: Childhood Autism Rating Scale; CBCL: Child Behavior Checklist; CPRS-R: Conners Parents Rating Scale-Revised; DIPAB: Danish Instrument of Psychotic Behavior in Autism; ECOS: Ecological Communication Orientation Scale; ERC: Evaluation Résumé du Comportement; GARS: Gilliam Autism Rating Scale; GFCF: Gluten Free: Casein Free; GSSI: Gastrointestinal Symptom Severity Index; ITPA: Illinois Test of Psycholinguistic Abilities; K-ABC: Kauffman Assessment Battery of Children; LIPS: Leiter International Performance Scale; MSEL: Mullen Scale of Early Learning; PDDBI: Pervasive Developmental Disorder Behavior Inventory; RFRLRS: Ritvo–Freeman Real Life Rating Scales; ROME III: Questionnaire for the assessment of functional gastrointestinal disorders; SCQ: Social Communication Questionnaire; Suppl.: supplements; TOMI: Test Of Motor Impairment; VABS: Vineland Adaptive Behavior Scale.

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
