# Peer review of "A Narrative Review about Autism Spectrum Disorders and Exclusion of Gluten and Casein from the Diet"

_nutrients, 2022, doi:10.3390/nu14091797_

Round 1

Reviewer 1 Report

The role of diet, especially the GFCF diet has been well-studied in ASD. A number of similar articles including a Cochrane review exist within the literature. Although an interesting topic, this article, as a narrative review does not add to the literature,

Author Response

We have cured our manuscript in order to improve it and considering the suggestions of all the other reviewers.

Reviewer 2 Report

The authors present a high quality, comprehensive critical review of the literature regarding use of gluten- and casein-free (GFCF) diet for the treatment of autism spectrum disorders (ASD).  Overall, I think the review is good, but there a few issues that should be addressed prior to publication:

  1. The quality of the writing is subpar, to the point that it impacts readability. There were even a few instances in which I did not understand what the authors were trying to convey.
  2. I think the review should focus more exclusively on GFCF diet. Sections 2 (microbiota-gut-brain axis in ASD) and 3.2 (nutritional supplements, prebiotics and probiotics) are more superficial and distract from the higher quality, more critical review of the GFCF literature. I would therefore remove those sections from the review. I would advise the authors not to write that they focus on GFCF diet because of "their previous background and knowledge about this topic" (Line 157-158). While this is obviously a practical consideration, readers will want a better justification for why they are reading about this topic, specifically.
  3. I agree with the authors' conclusion that there is insufficient evidence to recommend use of GFCF diet in ASD. However, instead of just presenting the positive studies and negatives, I would prefer the authors take a stronger stand on whether they think GFCF diet is effective. Personally, I think it is telling that the double blinded studies all have negative results.

Author Response

Thank you for the suggestion regarding English style. The manuscript has already been submitted to a thorough copyediting by an English-native speaker.

Regarding issue 2, both suggestions have been considered. We have deleted section 2 and subsection 3.2 as well as removed the sentence mentioning our previous background.

Finally, we have replaced "negative findings" by "lack of significant scientific evidence" to answer your last comment.

Reviewer 3 Report

Authors, in this narrative review, try to present their point of view on the current state of studies on GFCF diet in ASD.

In general, the Authors present the results of research on the use of GFCF diet in ASD in a balanced way. However, the section “4.2. Negative or inconclusive findings about GFCF diet” despite its title does not present any real negative outcomes of GFCF diet applied in ASD. Lack of adverse effect of either GFCF diet or supplementation of a diet with milk proteins and/or gluten is not equivalent to “negative results”. As negative results, one can expect, for example, a worsening in the nutritional status of the participants or autistic symptoms, or any parameter assessed in the studies (see e.g. DOI: 10.3390/nu13020470; https://doi.org/10.1007/s10803-015-2582-7). I recommend changing the title of the section and reformulating some sentences in it (see below) and/or clearly defining what effects of GFCF diet are considered negative by the Authors.

Minor remarks

Figure 2 – please update the information with the last two years

Table 1 last line in last column – please translate into English

Line 192 and later – “ScanBrit”

Line 224-232 – please reformulate the first sentence or specify “the negative side of gluten and casein removal from diet” in this section

Line 236 – what was the “negative results”? Did you mean lack of improvement? Please reformulate.

Line 244 – please reformulate the sentence. Lack of adverse effects is not “a negative result”

Author Response

Thank you very much for your accurate suggestion. We have reformulated all the sentences previously mentioning negative findings when they are actually related to the lack of any effect (negative or positive). These sentences are highlighted in yellow. In addition, we have incorporated the reference Marí-Bauset, et al.(2016) to the references list at line 162, as rightly mentioned the reviewer when a negative outcome is really obtained.

In that sense, we have also modified the title of section 3.2. Now it is “Non-significant improvement with GFCF diet”.

Regarding to minor remarks:

Figure 2 – please update the information with the last two years

This figure (currently Figure 1) has been updated with the last two years. In addition, we have added keywords and repository used (Pubmed).

Table 1 last line in last column – please translate into English

Done.

Line 192 and later – “ScanBrit”

“Scanbrit” has been replaced by ScanBrit, both origins with capital letters.

Line 224-232 – please reformulate the first sentence or specify “the negative side of gluten and casein removal from diet” in this section.

Done.

Line 236 – what was the “negative results”? Did you mean lack of improvement? Please reformulate.

Done.

Line 244 – please reformulate the sentence. Lack of adverse effects is not “a negative result”

Done.

Reviewer 4 Report

The paper is very interesting and very actual. I ask to the Authors to explain better the role of Microbiota at page 2 line 58-66 underlying the name of microbes involved and their role. At page 3, I would lke to have more informations about Ketogenic Diets

Author Response

According to suggestions performed by Reviewer 2, we have removed the section explaining the role of Microbiota as well as the section mentioning ketogenic diets.

Round 2

Reviewer 1 Report

The changes that have been made still leave the structure and information of the manuscript at an incomplete level.